# Bacterial Community Structure of *Pinus Thunbergii* Naturally Infected by the Nematode *Bursaphelenchus Xylophilus*

**DOI:** 10.3390/microorganisms8020307

**Published:** 2020-02-23

**Authors:** Yang Ma, Zhao-Lei Qu, Bing Liu, Jia-Jin Tan, Fred O. Asiegbu, Hui Sun

**Affiliations:** 1Collaborative Innovation Center of Sustainable Forestry in Southern China, College of Forestry, Nanjing Forestry University, Nanjing 210037, China; mayang0524@outlook.com (Y.M.); qzl941211@njfu.edu.cn (Z.-L.Q.); 202404324@163.com (B.L.); tanjiajin@njfu.edu.cn (J.-J.T.); 2Department of Forest Sciences, University of Helsinki, Helsinki 00790, Finland; fred.asiegbu@helsinki.fi

**Keywords:** pine wilt disease, bacterial community structure, functional structure, Illumina MiSeq sequence, PICRUSt analysis

## Abstract

Pine wilt disease (PWD) caused by the nematode *Bursaphelenchus*
*xylophilus* is a devastating disease in conifer forests in Eurasia. However, information on the effect of PWD on the host microbial community is limited. In this study, the bacterial community structure and potential function in the needles, roots, and soil of diseased pine were studied under field conditions using Illumina MiSeq coupled with Phylogenetic Investigation of Communities by Reconstruction of Unobserved states (PICRUSt) software. The results showed that the community and functional structure of healthy and diseased trees differed only in the roots and needles, respectively (*p* < 0.05). The needles, roots, and soil formed unique bacterial community and functional structures. The abundant phyla across all samples were Proteobacteria (41.9% of total sequence), Actinobacteria (29.0%), Acidobacteria (12.2%), Bacteroidetes (4.8%), and Planctomycetes (2.1%). The bacterial community in the healthy roots was dominated by Acidobacteria, Planctomycetes, and Rhizobiales, whereas in the diseased roots, Proteobacteria, Firmicutes, and Burkholderiales were dominant. Functionally, groups involved in the cell process and genetic information processing had a higher abundance in the diseased needles, which contributed to the difference in functional structure. The results indicate that PWD can only affect the host bacteria community structure and function in certain anatomical regions of the host tree.

## 1. Introduction

Plants in nature are closely associated with a variety of microorganisms. The surface and inner parts of the plant are occupied by different taxa of bacteria, fungi, archaea, and protists, which are collectively known as plant microbiota [1]. Plant microbial communities play important roles in host fitness. Some plant-related bacteria can have beneficial (symbiotic) effects, promoting plant growth and improving plant stress and disease resistance [2]. For example, *Paenibacillus polymyxa* can promote the growth of lodgepole pine seedlings via enhancing the nitrogen fixation of the root [3]. *Streptomyces* sp. ZL2 can increase the resistance of tomato to root rot caused by *Fusarium* sp. [4]. On the other hand, some bacteria can also have neutral, mutualistic, or negative (pathogenicity) interactions with host plants [5,6]. Some actinomycetes can colonize plant root internal tissues and form symbiotic relationships with plants to promote plant growth without causing disease symptoms [7].

Previous studies have shown that the decline in plant fitness (e.g., healthy status) or changes in growth conditions caused by host pathogens can affect the microbial community in leaves and roots of the host [8,9,10]. The healthy state of a plant can affect the exudates of the plant root, which is the essential factor affecting the rhizosphere bacterial community [11,12]. Therefore, it is important to elucidate the relationship between the plant pathogen and host microbial community.

Soil microbes, as decomposers, are involved in a series of complex biochemical reactions, including C, N, and P cycling [13]. Many factors, such as the soil pH, carbon to nitrogen ratio (C:N ratio), and water content, can influence the soil microbial community [14]. In a forest ecosystem, the forest disturbance and tree species can have direct or indirect effects on the soil microbial community via changing soil nutrient circulation [15,16]. The changes in the soil microbial community can in turn reflect the dynamics of soil quality and nutrient cycling [17,18]. Bacteria are the most abundant microorganisms in the soil, accounting for about 80% of the total soil microbes [15,19]. They are the early colonizers of easily available substrates and can transform energy from complex organic matter into forms that other organisms can easily absorb [20,21].

Pine wilt disease (PWD) caused by *Bursaphelenchus xylophilus* is one of the most serious conifer diseases worldwide, affecting several species of pine trees (*Pinus* spp.) and resulting in huge economic and environmental losses [22,23]. Many studies have focused on the pathogenicity mechanism of PWD [24,25], the biology of *B. xylophilus* [26,27], and the insect vectors [28,29]. Previous studies have shown that *B. xylophilus* infection can change the diversity and structure of endophytic wood-colonizing bacteria of *P*. *pinaster* trees and affect the abundance of certain taxonomic groups, e.g., *Streptomyces* and *Pseudomonas* [30]. However, the effect of PWD on the entire host microbial community under field conditions has been scarcely studied.

In this study, we hypothesized that the host bacterial community will change after the infection of *B. xylophilus*. Therefore, we selected *Pinus thunbergii* trees naturally infected by *B. xylophilus* and investigated the bacterial community structure and function in the needles, roots, and soil of diseased trees, using high-throughput Illumina MiSeq sequencing technology coupled with Phylogenetic Investigation of Communities by Reconstruction of Unobserved states (PICRUSt) functional analysis. The main aim of the study was to elucidate the shifts in the host microbial community and function caused by PWD and to better understand the relationships between pathogens and host microbial communities.

## 2. Materials and Methods

### 2.1. Study Sites and Sample Collection

The study site was located in The Sun Yat-sen Mausoleum Park in Purple Mountain, Nanjing, China (32°04′ N, 118°50′ E). The area has an annual average sunshine value of 1628.8 h, with an annual average temperature of 19.6 °C, annual average precipitation of 1530.1 mm, and frost-free period of 322 d. The area covers approximately 20 square kilometers, with mainly *P. thunbergii* Parl and *P. massoniana* Lamb forest aged around 70 years old, and Chinese *Cunninghamia lanceolata*. In addition, shrubs of *Symplocos paniculata*, *Camellia sinensis*, and *Lindera glauca*, and the herbal plants *Ophiopogon japonicus*, *Commelina communis*, and *Reynoutria japonica*, are common under the forest canopy. Three study plots with a distance of about 500 m apart and 20 × 20 m per plot were selected. In each plot, three *P. thunbergii* trees infected by *B. xylophilus* in March and killed in September, and three healthy trees, were selected for sampling. The distance between diseased and adjacent healthy trees was less than 15 m. The selection of diseased and healthy trees was made according to the method described by Millberg et al. [31], in which the healthy tree refers to needles that are completely green and the diseased tree refers to needles that have died with a dry and brownish symptom of PWD. The subsequent confirmation of healthy and diseased trees was carried out in the laboratory by isolation of the nematode and using specific primers of *B. xylophilus* [32].

For the sampling, after each tree was felled, 15 needles from the tips of the shoot in the middle of the canopy from three directions, including 120° around the tree, were collected and mixed as one sample. The root samples were obtained by a sterilized punch (diameter 10 mm) in the roots about 25 cm underneath the soil surface in three directions (120° as the boundary) from each tree and mixed as one sample. In total, 18 needle and 18 root samples (9 diseased and 9 healthy) were obtained. The soil samples were collected after litter removal around each selected tree, with a distance of about 20, 40, and 60 cm from the tree in each of the three directions (120° as the boundary). Due to the heterogeneity variations in the soil being much larger than those in the needles and roots, three samples in one direction for each tree were pooled together as one replication, resulting in three biological replicates per a tree. A total of 54 soil samples (3 samples/tree × 3 trees/plot × 3 plots × 2 (diseased and healthy trees)) were obtained. All the samples were put in the ice box and transported to the laboratory for subsequent analysis.

### 2.2. Soil Property Analysis

The soil water content (SWC) was measured by a weighting method after drying the soil [33]. The soil pH was measured in a 1:2.5 suspension [34]. The soil organic matter (SOM) was calculated from the percent of organic carbon determined by the K_2_Cr_2_O_7_ wet combustion method [35]. The total nitrogen (TN) was measured following the Kjeldahl method [36]. The Microbial Biomass Carbon (MBC) was calculated with the extraction coefficient of 0.38 following chloroform fumigation extraction [37].

### 2.3. DNA Extraction, Amplification of the 16S rDNA Region, and Illumina MiSeq Sequencing

The soil genomic DNA was extracted from 0.3 g (fresh weight) homogenized soil using a Soil DNA kit (OMEGA BIO TEK, Norcross, GA, USA), following the manufacturer’s instructions. The genomic DNA from the needles and roots was extracted using a Plant Genomic DNA Kit (TIANGEN BIOTECH (BEIJING) CO., LTD, Beijing, China city, state abbreviation if USA or Canada, country). The concentrations of DNA were measured using a NanoDrop ND-1000 spectrophotometer (Thermo Fisher Scientific, USA). The DNA was subjected to polymerase chain reaction (PCR) amplification of the V3–V4 region of the bacterial 16S rDNA gene using primer pairs of 338F (ACTCCTACGGGAGGCAGCAG) and 806R (GGACTACHVGGGTWTCTAAT) [38] containing partial adapter sequences at the 5′ ends. The TransGen AP221–0220 µL reaction system was used in this experiment. The reaction included 4 µL 5× FastPfu Buffer, 2 µL dNTPs (2.5 mM), 0.8µL Forward Primer (5 µM), 0.8µL Reverse Primer (5 µM), 0.4 µL FastPfu Polymerase, 0.2 µL BSA, and 10 ng Template DNA. The PCR reaction parameters were as follows: 95 °C for 3 min, 27 cycles of 95 °C for 30 s, an annealing temperature of 55 °C for 30 s, 72 °C for 45 s, and then a final extension of 10 min at 72 °C. A negative PCR with sterilized water as a template was included to track the possible contaminations. The PCR product was detected using 2% agarose gel electrophoresis and purified with Agencourt AMPure XP kit (Beck-man Coulter Life Sciences, IN, USA). The concentration was measured using a Nano-drop ND-1000 spectrophotometer and was subjected for sequencing with paired-end (PE = 300) Illumina MiSeq platform at Majorbio (Shanghai International Medical Zone, Shanghai, China). Raw sequences were deposited at the Sequence Read Archive (SRA) of the National Center for Biotechnology Information (NCBI) under project accession number PRJNA577573.

### 2.4. Sequence Data Processing, Information on Illumina MiSeq Data, and Statistical Analysis

The raw sequence data were processed using Mothur software [39] following the Standard Operating Procedure (SOP). Adapter and barcode sequences were removed using Cutadapt v.1.15 [40]. The pre-processed sequences were quality checked for sequencing errors (trim.seqs), PCR errors (pcr.seqs), and chimera (chimera.uchime) using Mothur commands. The sequences were aligned against the SILVA database (reference = silva.nr_v132) with the align.seqs command. The sequences were then pre-clustered with 6 bp differences (pre.cluster) and clustered to form operational taxon units (OTUs) at 97% similarity. OTUs with sequence numbers of less than 10 among all samples were screened out [41]. The sequences were assigned to a taxonomic group with an 80% bootstrap confidence by using the RDP Naïve Bayesian rRNA Classifier tool version 2.0 [42]. Sequences assigned to the plant chloroplast and non-bacteria domain were filtered out.

A total of 2,120,302 high-quality sequences were generated across all samples after sequence de-noising and quality filtering. The average number of sequences per sample was 23,559 ± 10,893 (mean ± standard deviation), and ranged from 2350 to 54,879 per sample. For data normalization, 2350 sequences were randomly subsampled from each sample for calculating the diversity index and community structure comparison.

The community richness (Sobs and Chao1), diversity (Shannon), and evenness (Shannoneven) were calculated by subsampled data with the smallest size of the sample across all samples. One-way analysis of variance (ANOVA) tests were used to identify differences in community richness, diversity, and evenness among treatments. Venn diagrams were constructed using subsampled data to show shared and unique OTUs with InteractiVenn (http://www.interactivenn.net) [43]. Linear discriminant analysis (LDA) coupled with effect size (LEfSe; http://huttenhower.sph.harvard.edu/galaxy/root?tool_id=PICRUSt_normalize) was used to identify the bacterial taxonomic and functional groups differentially represented between treatments [44]. The criterion for LEfSe was set as LDA > 3.5 with *p* < 0.05. Principal co-ordinates analysis (PCoA) was used to visualize the bacterial community structure and canonical correspondence analysis (CCA) was used to visualize the bacterial functional structure with Bray–Curtis similarity using relative abundances of OTUs or functional groups in PRIMER v.7 [45], with the add-on package of PERMANOVA^+^ [46]. Prior to PCoA and CCA, the data were square-rooted transformed to meet the analysis criteria. Subsequently, a PERMANOVA test was used to determine the significant difference in community structure between treatments.

Phylogenetic Investigation of Communities by Reconstruction of Unobserved states (PICRUSt) software was used to predict the community potential function (http://picrust.github.io/picrust) [47], which contains six functional groups, including Metabolism, Environmental Information Processing, Genetic Information Processing, Cellular Processes, Human Diseases, and Organismal Systems. In each group, the potential function was further assigned to a second level with more subgroups.

## 3. Results

### 3.1. Soil Physical and Chemical Properties Around Diseased and Healthy Trees

The results of soil property analysis showed that the soil total nitrogen content, soil pH, and microbial biomass carbon differed between the diseased and healthy soils, and were significantly higher in soils surrounding diseased trees (*p* < 0.05) (Table 1). The soil water content and soil organic matter did not differ between the two soils.

### 3.2. Bacterial Community Diversity between Diseased and Healthy Trees

The community richness and diversity in the soil, roots, and needles of diseased samples were all higher than in healthy samples, respectively (Appendix A). However, no significant differences were observed between the healthy and diseased samples for any of the diversity indices. In addition, the soil had the highest community richness, diversity, and evenness in both healthy and diseased trees, followed by the roots and needles. The bacterial community diversity indices significantly differed among the soil, roots, and needles (*p* < 0.05 for all pairs).

### 3.3. Bacterial Community Structure at the Taxonomic Level between Diseased and Healthy Trees

All sequences were classified to the bacterial domain and assigned to 15,130 OTUs across all samples, including 11 bacterial phyla, 57 classes, 99 orders, and 462 genera. Proteobacteria (41.9% of the sequences and 33.4% of the OTUs) was the most abundant phylum, followed by Actinobacteria (29.0% of the sequences and 17.0% of the OTUs), Acidobacteria (12.2% of the sequences and 13.1% of the OTUs), Bacteroidetes (4.8% of the sequences and 7.7% of the OTUs), Planctomycetes (2.1% of the sequences and 5.0% of the OTUs), Verrucomicrobia (1.8% of the sequences and 3.2% of the OTUs), Candidatus Saccharibacteria (1.4% of the sequences and 4.9% of the OTUs), and Chloroflexi (1.2% of the sequences and 2.2% of the OTUs) (Figure 1). The less abundant phyla (<1.0% of the sequences) included Genmmatimonadetes (0.6% of the sequences and 1.2% of the OTUs) and Firmicutes (0.5% of the sequences and 1.2% of the OTUs) (Appendix A).

At the order level, Actinomycetales was the most abundant order (16.5% of the sequences and 8.8% of OTUs), followed by Rhizobiales (13.9% of the sequence and 5.3% of OTUs), Rhodospirillales (6.1% of the sequence and 5.4% of OTUs), Burkhoderiales (5.0% of the sequence and 3.0% of OTUs), Gaiellales (4.0% of the sequence and 2.0% of OTUs), Sphingomonadales (3.5% of the sequence and 1.6% of OTUs), and Gp6 (3.0% of the sequence and 1.5% of OTUs). At the genus level, the abundant genera included *Gaiella* (4.0% of the sequence and 2.0% of OTUs), *Burkholderia* (2.3% of the sequence and 0.8% of OTUs), *Sphingomonas* (1.7% of the sequence and 0.5% of OTUs), *Massilia* (1.4% of the sequence and 0.2% of OTUs), *Mycobacterium* (1.2% of the sequence and 0.6% of OTUs), *Actinospica* (1.1% of the sequence and 0.2% of OTUs), and *Methylobacterium* (1.0% of the sequence and 0.1% of OTUs). A detailed list of the orders and genera is shown in Appendix A.

The Lefse analysis showed that the abundance of some taxa differed between the healthy and diseased samples in the needles, roots, and soil, respectively (LDA >3.5, *p* < 0.05). In the needles, the phylum Candidatus Saccharibacteria, the order Burkhoderiales, and the genera *Massilia* were more abundant in the diseased tree, whereas the order Rhizobiales and the genus *Beijerinckia* had a higher abundance in the healthy tree (Figure 2a). In the roots, the phyla Proteobacteria and Firmicutes; the orders Burkhoderiales, Sphingomonadales, Enterobacteriales, and Xanthomonadales; and the genera *Burkholderia*, *Novosphingobium*, *Pseudomonas*, and *Sphingomonas* were more abundant in the diseased tree, whereas the phyla Acidobacteria, Planctomycetes, and Chloroflexi; the orders Rhizobiales, Planctomycetales, Gp1, and Gp 2; and the genera *Aquisphaera* had a higher abundance in the healthy tree (Figure 2b). In the soil, no difference was found in the abundance at a phylum level between healthy and diseased trees and only the genus *Bradyrhzobium* was more abundant in soil surrounding the diseased tree (Figure 2c).

### 3.4. Bacteria Community Structure at an Operational Taxon Unit (OTU) Level between Diseased and Healthy Trees

The number of shared and unique OTUs between healthy and diseased samples in the needles, roots, and soil differed (Figure 3a–c). The soil shared half of the OTUs (50.5%) between diseased and healthy samples, followed by the needles (32.5%) and roots (23.7%). Only 0.7% of the OTUs was shared among the needles, roots, and soil, in which the soil harbored the most unique OTUs, followed by the roots and needles (Figure 3d–e).

PCoA analysis based on the OTU data detected 29.6% of the total variance among bacterial communities, with the first and second axes explaining 22.9% and 12% of the variance, respectively (Figure 4). The needles, roots, and soil formed distinct bacterial communities, with subsequent PERMANOVA confirming the significance among them (*p* < 0.05 in all possible pairs). The difference in the bacterial community between the healthy and diseased samples was only detected in the roots, and not in the needles and soil. The top 13 OTUs significantly contributed to the community shift in the roots, in which *Burkholderia* (OTU000485 and OTU000403), *Novosphingobium* (OTU006708), Rhizobiales (OTU000236), Bradyrhizobiaceae (OTU00245 and OTU000397), and Actinomycetales (OTU8340) had a higher abundance in diseased roots (*p* < 0.05), whereas, *Roseiarcus* (OTU00384), *Thermus* (OTU001566), Rhizobiales (OTU001498), Bradyrhizobiaceae (OTU000317 and OTU000184), and Thermomonosporaceae (OTU001381) had a higher abundance in healthy roots (*p* < 0.05) (Table 2). The bacterial community structure did not differ between the healthy and diseased trees in the needles and soil; however, some OTUs in either the needles or soil showed significant differences in the abundance. The abundances of the top 13 OTUs (>0.04%) in the needles, roots, and soil are shown in Table 2.

### 3.5. Bacterial Community Structure of the Predicted Function between Diseased and Healthy Trees

A total of 788 OTUs (5.9% of the total) were matched to the reference Greengene database and included in the PICRUSt analysis for the predicted function. The OTUs were assigned to six functional groups and 41 sub-groups. Of these, the group Metabolism had the highest abundance (51.1%), followed by Environmental Information Processing (15.5%), Genetic Information Processing (14.4%), Cellular Processes (3.6%), Human Diseases (1.1%), and Organismal Systems (0.8%). The Lefse analysis showed that the abundance of some functional groups differed in abundance between the diseased and healthy samples in the needles, roots, and soil, respectively (LDA > 3, *p* < 0.05). In the needles, the abundance of Cellular Processes (at a group level), and the Replication and Repair, Cellular Processes and Signaling, and Cell Motility (at a sub-group level) were higher in diseased needles, whereas the Environmental Information Processing and Signaling and Signal Transduction (at a group level), and the Membrane Transport and Xenobiotics Biodegradation and Metabolism (at a sub-group level) had a higher abundance in healthy needles (Figure 5a). The Metabolism (at a group level) and Xenobiotics Biodegradation (at a sub-group level) were more abundant in diseased roots and soil, and the Membrane Transport (at a sub-group level) had a higher abundance in diseased soil (Figure 5b,c). The CCA analysis showed that only the healthy and diseased needles formed different bacterial functional structures (*p* < 0.05), and not the roots or soil (Figure 6).

## 4. Discussion

We investigated the bacterial community in needles, roots, and soil from healthy and diseased pine trees naturally infected by *B. xylophilus* under field conditions. As a previous study showed that the infection of plant pathogens can affect the host microbial community [8,9], we expected the pine wilt disease to have a profound impact on the host bacterial community structure. However, we did not observe any significant difference in community diversity between the healthy and diseased trees in needle, root, or soil samples. Similar results have been observed in Norway Spruce infected by root rot pathogen *Heterobasidion* sp., in which the host microbial diversity was similar for symptomatic and asymptomatic trees in different anatomic regions of the host [48]. The colonization of microorganisms in the phyllosphere is regulated by the stomata and microbes in the needles/leaves seem to be more susceptible to precipitation factors than diseases [49]. In our study, as well as the previous one [48], the endophyte community was not investigated. In addition, fungi are the main decomposers of litter on the soil surface in pine forests, while the importance of bacteria increases with the increase of soil depth [50]. The soil samples in this study were taken from the surface, which may partly explain the results of diversity, with no differences between soil from areas surrounding healthy and diseased trees being exhibited. On the contrary, Proença et al. [30] found that PWD can increase the diversity of endophytic wood-colonizing bacteria in the trunk of *P*. *pinaster* trees and the endophytic bacterial community differed as the disease progressed, suggesting the importance of disease development in the host microbial community. There can be a few months between the occurrence of initial symptoms after *B. xylophilus* infection and tree death. Unfortunately, we did not collect samples from the trunks of trees in our study. The samples were only collected at one time point at the last stage of the disease and were collected at multiple time points following PWD development. Therefore, it is necessary to further investigate the bacterial community at different stages of PWD and in the soil at different depths after PWD occurrence.

In this study, the community structure of healthy and diseased trees was only significantly different in the roots, and not in the needles and soil. Previous studies have shown that root exudates are the essential factor determining the structure of the root and rhizosphere microbial community [51,52,53]. The occurrence of pine wilt disease can lead to a decreased secretion of soluble sugar, total sugar, and protein in roots [54], which might have caused the observed difference in the microbial community structure in the roots. In addition, the changes in the ectomycorrhizal fungi (ECMF) structure of *Pinus tabulaeformis* caused by pine wilt disease has previously been reported [55]. Therefore, it is most likely that pine wilt disease will significantly affect the microbial community structure in the roots and rhizosphere of *P. thunbergii*. As we only collected the main root of the host in our study, further investigations of the microbial community in the rhizosphere of soil and fine roots of the host are necessary to validate our findings and assumptions. New infections of the pine wilt nematode usually occur in March and the infected tree will die after a few months (around September) when there is less precipitation. The period from the appearance of symptoms to tree death can be very short (2–4 weeks) [56]. In this study, samples were collected from dead trees in October. The timing of sample collection might have impacted the detection of changes in the microbial community in the needles and soil. In the roots, the relative abundances of Proteobacteria and Firmicutes were significantly higher in diseased roots. Proteobacteria can be phytopathogens and parasites in plant tissues and cause a variety of diseases [57]. The root metabolism of diseased trees was weakened compared to the healthy roots, resulting in a decreased ability of the root to adapt to the soil condition and it being easily colonized by microbes. Proteobacteria prefer to grow in nutrient-rich conditions [56], which may explain the high content of Proteobacteria in the diseased roots. The response and oxygen content in the roots will decrease after the tree dies. This might cause a higher abundance of Firmicutes in diseased roots as it prefers anaerobic environments [58]. The genera *Burkholderia* and *Pseudomonas* also had a higher abundance in diseased roots. Both genera are gram-negative bacteria that are widely found in water, soil, and plants [59,60]. Studies have shown that *Burkholderia* can cause onion stem rot, and *Pseudomonas* can cause root rot of *Arabidopsis thaliana* [61]. The high abundance in diseased roots may have been intended to increase root decay.

Although the bacterial community structure did not differ between healthy and diseased samples in the needles and soil, certain taxa shifted in abundance. The abundances of the phylum Candidatus Saccharibacteria and genus *Massilia* in diseased needles were significantly higher than those in healthy needles. A previous study has shown that Candidatus Saccharibacteria has the ability to degrade cellulose [62]. The higher abundance of Candidatus Saccharibacteria in diseased needles may accelerate the decomposition of cellulose in the needles. The genus *Massilia* belongs to the family Oxalobacteraceae of the class Betaproteobacteria in the phylum Proteobacteria [63]. Members of this genus are characterized as Gram-negative, aerobic, non-spore-forming bacteria [64]. Some *Massilia* can produce cell lysis enzymes that promote tissue lysis [65]. This may be the reason for the presence of *Massilia* in a high abundance in diseased needles. In the soil, only the genus *Bradyrhzobium* had a significantly higher abundance in diseased than healthy soil. The elevated soil pH in PWD-infested soil could be partly because *Bradyrhzobium* prefers acidic soils [66].

The function prediction analysis of PICRUSt based on high-throughput sequencing has been applied in analyzing the ecological function of different plants [67,68]. We hypothesized that changes in the bacterial community structure would elicit a functional response. Differences in the bacterial community structure between healthy and diseased roots were observed; however, no difference was found in the bacterial functional structure, which may suggest that there was no direct correlation between changes in the bacterial community and functional structure induced by PWD. However, PICRUSt analysis with only small portion of the OTUs included surely limited the interpretation of the results. The functional prediction was based on the classification of OTUs and the Greengene database as a reference. The result would have been more reliable with a high number of lower taxonomic level OTUs (e.g., genera or species) or a more accurate reference database. In our study, the number of OTUs classified at a genus level was low (24.6%), resulting in only 788 OTUs matching the Greengene database. Further investigation is needed, in combination with more powerful tools, e.g., metagenomic sequencing and other technologies, to elucidate the microbial community function.

The soil pH, soil TN, and microbial biomass carbon in diseased soil increased significantly. Previous studies have shown that PWD significantly increased soil TN and microbial biomass carbon in a *P. tabulaeformis* forest [55,69]. Conifer species can secrete organic acids and absorb basic cations through their mycorrhizal associations, thus promoting soil acidification [70,71]. The mycorrhizal biological action ceased under the influence of PWD, resulting in a significant increase in the pH of diseased soil. In addition, due to needle shedding after PWD, the increase in light and temperature in the forest, to some extent, can promote the growth of microorganisms and decomposition of surface litters, change the transformation properties of soil nitrogen, and increase the mineralization rate of soil nitrogen [72].

## 5. Conclusions

In conclusion, PWD did not affect the host and soil bacterial community diversity. The differences in bacterial community structure and function between healthy and diseased trees were only observed in roots and needles, respectively, suggesting that PWD can only affect the host bacteria community structure and function in certain anatomic regions of the host tree. The bacterial community in the healthy root was dominated by Acidobacteria (phylum), Planctomycetes (phylum), and *Aquisphaera* (genus), whereas, in diseased root, Proteobacteria (phylum), Firmicutes (phylum), *Burkholderia* (genus), and *Pseudomonas* (genus) were dominant. In addition, different taxonomic regions (needles and roots) of the host harbored unique bacterial communities. Functionally, the groups involved in cell process and genetic information processing had a higher abundance in the diseased needles, which contributed to the differences in the functional structure. Moreover, PWD can change certain bacterial taxonomic and functional groups in investigated regions of the host, despite unchanged bacterial structures. Further investigation of the host microbial community and function in different stages of PWD and in the rhizosphere is needed to elucidate the effect of PWD on the host plant microbiome.

## Figures and Tables

**Figure 1 microorganisms-08-00307-f001:**
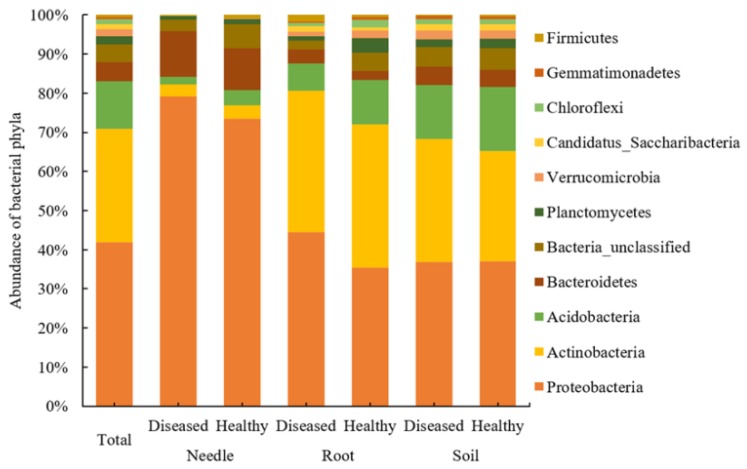
The relative abundance of bacterial phyla (% of the total number of reads) in healthy and diseased samples in the needles, roots, and soil of *Pinus thunbergii*.

**Figure 2 microorganisms-08-00307-f002:**
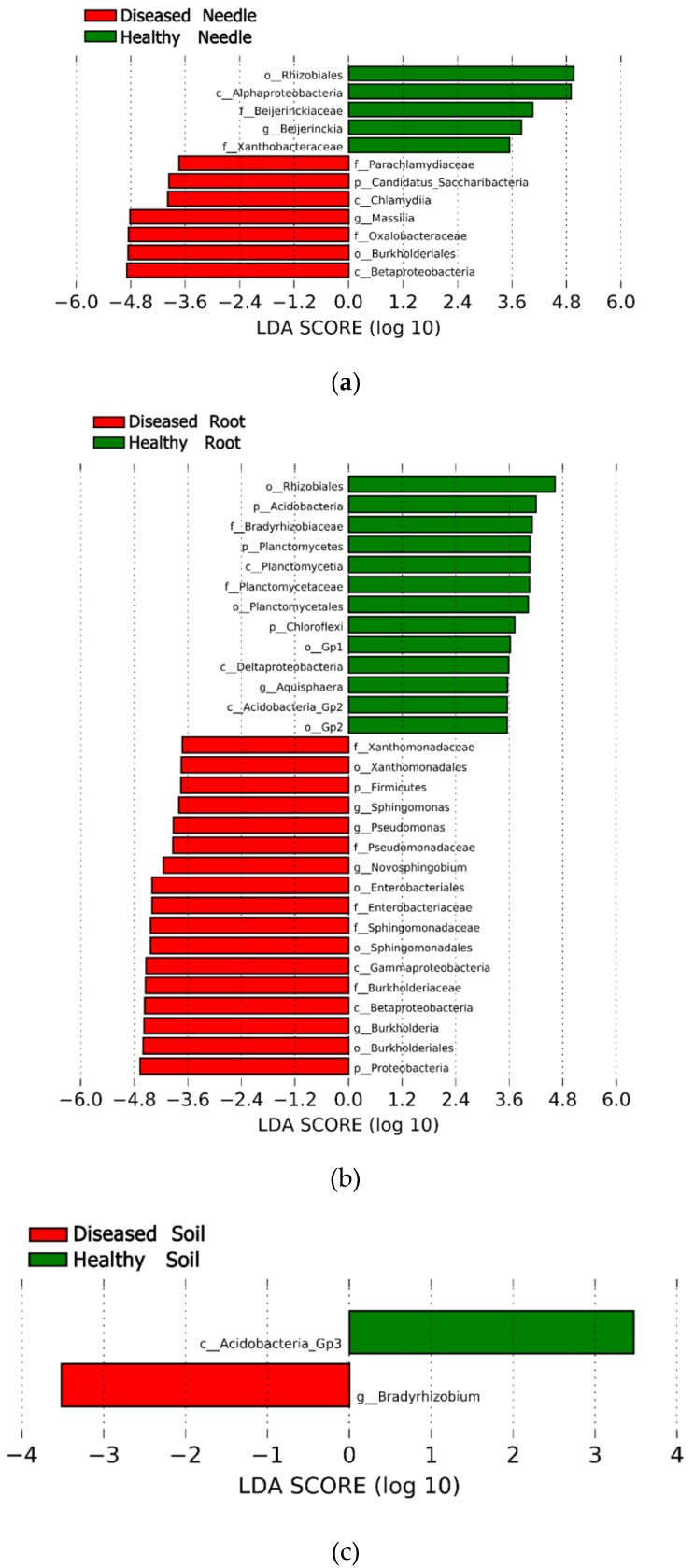
Lefse analysis showing the significant differences at different bacterial taxonomic levels between diseased and healthy samples in the needles (**a**), roots (**b**), and soil (**c**). Abbreviation: p: phylum, c: class, f: family, o: order, and g: genus.

**Figure 3 microorganisms-08-00307-f003:**
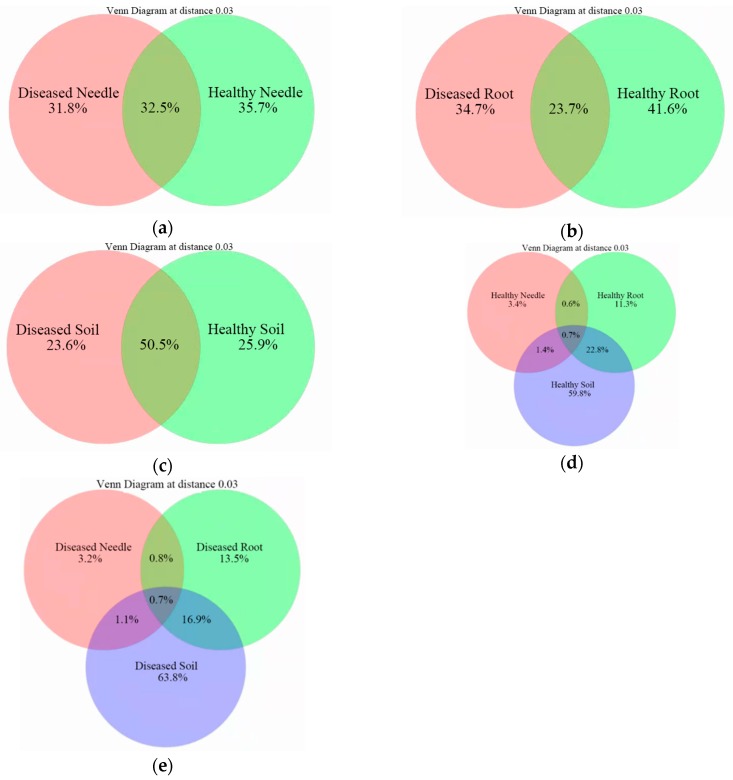
Venn diagram showing the unique and shared operational taxon units (OTUs) between healthy and diseased samples in the needles (**a**), roots (**b**), and soil (**c**), and among the needles, roots, and soil in healthy (**d**) and diseased (**e**) samples.

**Figure 4 microorganisms-08-00307-f004:**
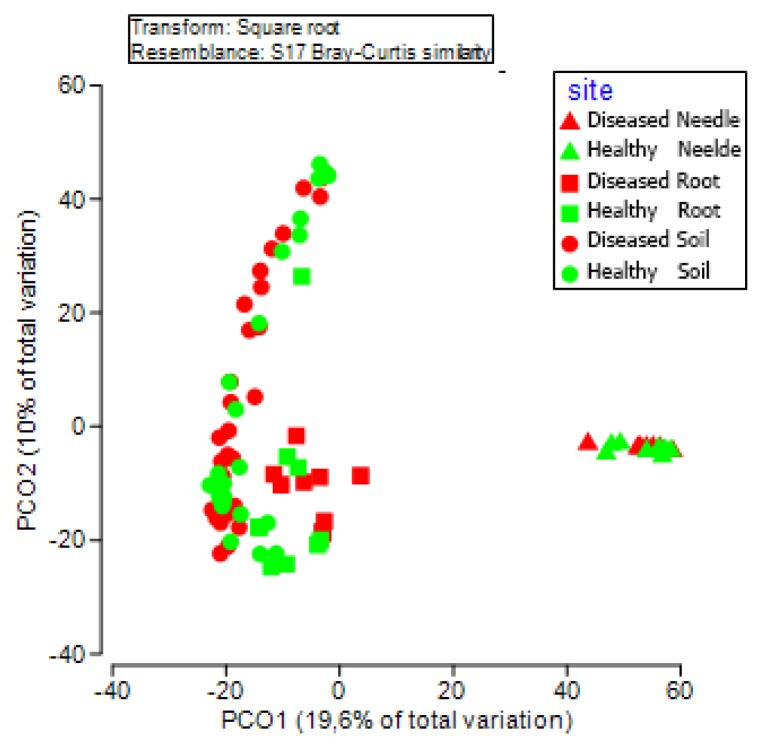
Principal co-ordinates analysis (PCoA) showing the bacterial community structure in healthy and diseased samples in the needles, roots, and soil.

**Figure 5 microorganisms-08-00307-f005:**
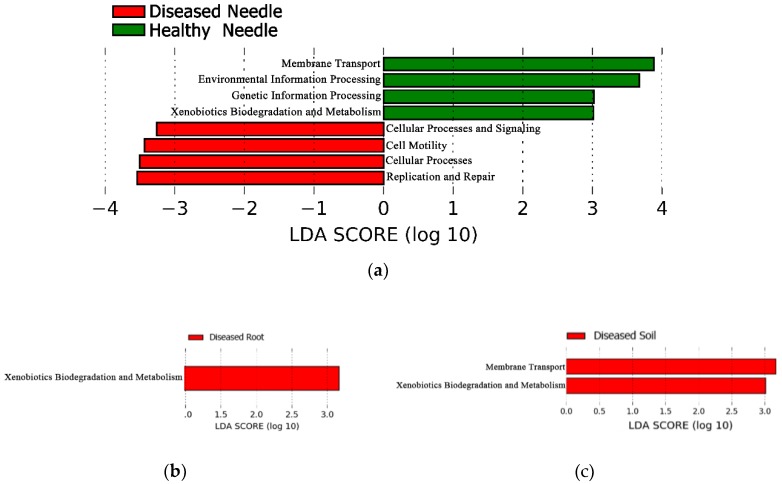
Lefse analysis showing the predicted functional groups significantly presented for healthy and diseased samples in the needles (**a**), roots (**b**), and soil (**c**).

**Figure 6 microorganisms-08-00307-f006:**
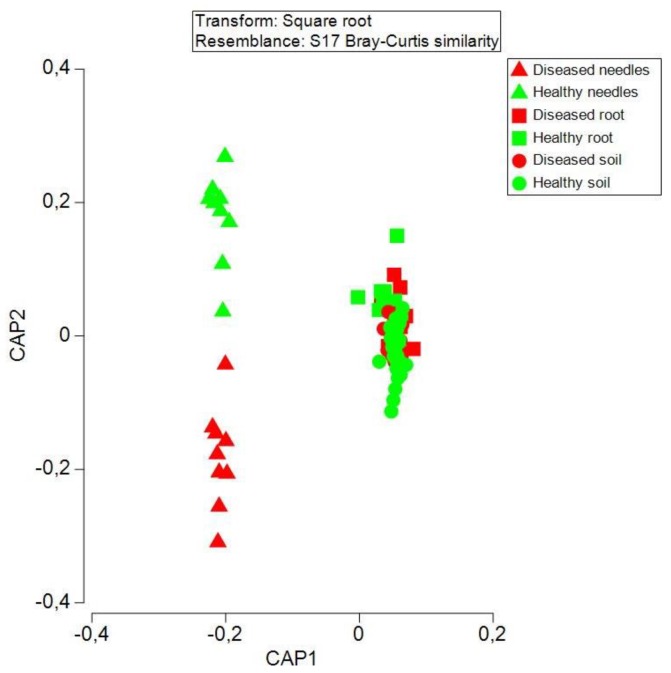
Canonical correspondence analysis (CCA) showing the bacterial functional structure in healthy and diseased samples in the needles, roots, and soil.

**Table 1 microorganisms-08-00307-t001:** The physical and chemical properties in healthy and diseased soil.

Soil	SWC	SOM (g/kg)	pH	MBC	TN (g/kg)
Diseased soil	0.32 ± 0.04	149.72 ± 6.70	5.57 ± 0.08 *	232.45 ± 5.94 *	21.74 ± 0.62 *
Healthy soil	0.30 ± 0.01	141.49 ± 8.24	5.16 ± 0.01	170.38 ± 0.94	19.28 ± 0.53

The values are shown as means ± standard deviation (*n* = 27). * *p* < 0.05, significant difference in the global Kruskal–Wallis test. SWC: soil water content; SOM: soil organic matter; MBC: microbial biomass carbon; TN: total nitrogen.

**Table 2 microorganisms-08-00307-t002:** The list of the top 13 (>0.04%) OTUs showing significant differences in the abundance between healthy (H) and diseased (D) trees in the roots (R), needles (N), and soil (S).

Site	OTUs	Taxonomy	*p* Value	Abundance Pattern
Root	Otu000485	*Burkholderia*	0.008991	HR < DR
	Otu000403	*Burkholderia*	0.003996	HR < DR
	Otu006708	*Novosphingobium*	0.000999	HR < DR
	Otu000236	Rhizobiales	0.000999	HR < DR
	Otu000245	Bradyrhizobiaceae	0.000999	HR < DR
	Otu000397	Bradyrhizobiaceae	0.000999	HR < DR
	Otu008340	Actinomycetales	0.000999	HR < DR
	Otu000384	*Roseiarcus*	0.012987	HR > DR
	Otu001566	*Thermus*	0.002997	HR > DR
	Otu001498	Rhizobiales	0.038961	HR > DR
	Otu000317	Bradyrhizobiaceae	0.004995	HR > DR
	Otu000184	Bradyrhizobiaceae	0.000999	HR > DR
	Otu001381	Thermomonosporaceae	0.000999	HR > DR
Needle	Otu001083	*Massilia*	0.000999	HN < DN
	Otu003662	*Massilia*	0.000999	HN < DN
	Otu000367	*Sphingomonas*	0.000999	HN < DN
	Otu001020	*Sphingomonas*	0.000999	HN < DN
	Otu001413	*Novosphingobium*	0.010989	HN < DN
	Otu006094	*Hymenobacter*	0.000999	HN < DN
	Otu005175	*Hymenobacter*	0.003996	HN < DN
	Otu006843	Proteobacteria	0.008991	HN < DN
	Otu002268	Burkholderiales	0.000999	HN < DN
	Otu004298	Sphingobacteriaceae	0.027972	HN < DN
	Otu000064	*Sphingomonas*	0.000999	HN > DN
	Otu002391	Rhizobiales	0.018981	HN > DN
	Otu001673	Acetobacteraceae	0.012987	HN > DN
Soil	Otu000475	*Gaiella*	0.042957	HS < DS
	Otu001002	*Gaiella*	0.038961	HS < DS
	Otu001492	*Sphingomonas*	0.041374	HS < DS
	Otu000195	Actinobacteria	0.038961	HS < DS
	Otu000376	Acidimicrobiales	0.027972	HS < DS
	Otu001968	Acidimicrobiales	0.023976	HS < DS
	Otu000482	Rhizobiales	0.02997	HS < DS
	Otu000313	Rhodospirillales	0.000999	HS < DS
	Otu000184	Bradyrhizobiaceae	0.03996	HS < DS
	Otu002791	Thermomonosporaceae	0.035964	HS < DS
	Otu000288	Rhizobiales	0.005994	HS > DS
	Otu000706	Rhizobiales	0.000999	HS > DS
	Otu001478	Actinomycetales	0.004546	HS > DS

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
