# Peer review of "Bacterial Community Structure of Pinus Thunbergii Naturally Infected by the Nematode Bursaphelenchus Xylophilus"

_microorganisms, 2020, doi:10.3390/microorganisms8020307_

Round 1
Reviewer 1 Report
The manuscript deals with the analysis of bacterial community structure and potential function in needle, root and soil of Pinus thunbergii infected by the nematode Bursaphelenchus xylophilus under field condition. Overall, the paper presents many weak points that limit its quality. In particular, it is not clear the relevance of studying changes in bacterial community structures of Pinus thunbergii infected by the nematode. The introduction should be more focused on this aspect and should be improved.
The significance of the results is not clearly reported. The Authors state that “PWD had effect on host bacteria community structure and function in the anatomic regions of the host tree” without providing any hopothesis of mechanism. Moreover, the relationship between nematode infection and host microbial communities was not mentioned and discussed.
Another weak point concerns the sample size which is different between needle and roots (18 samples each) and the soil (54 samples). The Author should give a convincing explanation.
Specific points:
Supplementary tables are difficult to read as the rows are often not aligned properly. The font in figures 2b, 3 and 5 is too small. Lines 193-202: there is no correspondence between the text and figure 2 a, 2b and 2c. The microbial taxa mentioned in the text are not highlighted in the figure. Lines 232-233: the Authors should explain the reason why they have chosen to analyze only 5.9% of total OTUs. Indeed, this limits the interpretation of the results, as reported by the Authors in the Discussion (lines 311-312). The list provided in Table 2 refer to OTUs showing significant differences in abundance between healthy and diseased trees in root, needle and soil. These informations, however, are discordant with respect to those reported in Tables S2 and S3. For example, in roots Burkholderiales appeared to be significantly different in Table S3 but they are absent in Table 2. The same applies for Sphingomonadales and Sphingomonas, etc.
Minor points:
Line 49: delete “Soil” Line 63: change “is” to “was” Line 301: delete “-stain”Author Response
Response to Reviewer 1 Comments
Comments and Suggestions for Authors
The manuscript deals with the analysis of bacterial community structure and potential function in needle, root and soil of Pinus thunbergii infected by the nematode Bursaphelenchus xylophilus under field condition. Overall, the paper presents many weak points that limit its quality.
Thank you so much for the valuable comments. We have revised the manuscript thoroughly according to the comments.
Point 1: In particular, it is not clear the relevance of studying changes in bacterial community structures of Pinus thunbergii infected by the nematode. The introduction should be more focused on this aspect and should be improved.
Response 1: Thank you for the insightful comments. The importance to elucidate the relationship between the plant pathogen and host microbial community was addressed in the introduction. We have added the previous studies showing the effect of the infection of pathogens on the host microbial community in the introduction (line 41-53, line 67-70), based on which we formed our hypothesis (L77-78).
Point 2: The significance of the results is not clearly reported. The Authors state that “PWD had effect on host bacteria community structure and function in the anatomic regions of the host tree” without providing any hopothesis of mechanism. Moreover, the relationship between nematode infection and host microbial communities was not mentioned and discussed.
Response 2: Thanks for the comments and pointing out the mistake. The reviewer is right that the sentence was not properly stated as PWD affected the bacterial structure only in the root and changed the functional structure only in the needle. We have corrected and stated more precisely in the text as ‘PWD can affect the host bacteria community structure and function only in certain anatomic regions of the host tree’.
For the results, we slighted changed the structure and put the ‘information on Illumina MiSeq data’ in the section of Materials and methods and highlighted the sub-titles in the each section. We focused on the comparison beween diseased and healthy trees on soil properties (3.1), diversity (3.2), structure at taxonomic level (3.3) and OTUs level (3.4), and structure at functional level (3.5).
Ponit 3: Another weak point concerns the sample size which is different between needle and roots (18 samples each) and the soil (54 samples). The Author should give a convincing explanation.
Response 3: Thanks for the comments. We collected one mixed sample of needl or root from each tree to minimize the variation between samples. However, the heterogeneity variations in the soil are much larger compared to the needle and root. Therefore, to minimize and compensate the possible variations, we collected three mixed samples from each tree instead of one mixed sample, resulting different sample size to the sample number in the needle and root. Secondly, our main focus is to compare the differences between diseased and healthy sample within needle, root or soil, respectively, but not among them. We have added explanation for the difference in soil sample size.
Specific points:
Ponit 4: Supplementary tables are difficult to read as the rows are often not aligned properly.
Response 4: Thanks for the comments. We have revised the Supplementary tables according to the comments.
Point 5: The font in figures 2b, 3 and 5 is too small. Lines 193-202: there is no correspondence between the text and figure 2 a, 2b and 2c. The microbial taxa mentioned in the text are not highlighted in the figure.
Response 5: Thanks for the comments. We have reconstructed the Figure 2, Figure 3 and Figure 5. The text has been rewritten based on the Figure 2, in which the plylum and genera mentioned were highlighted
Point 6: Lines 232-233: the Authors should explain the reason why they have chosen to analyze only 5.9% of total OTUs. Indeed, this limits the interpretation of the results, as reported by the Authors in the Discussion (lines 311-312).
Response 6: The functional prediction was based on the the classification of OTUs from Miseq. data and the Greengene database as reference, in which the OTUs with the similarity to the "closed reference" in reference database (the default similarity is ≥ 97%) will be selected and further classified to functional level. The result will be more reliable if the number of OTUs with lower taxonomic level (e.g. genus or species) were higher and the reference database is more accurate. In our study, the number of OTUs classified to genus level was low (24.6%), resulting only 788 OTUs matching in the Greengene database, resulting low number of OTUs included in the analysis. We have revised the text in the discussion based on the comments.
Point 7: The list provided in Table 2 refer to OTUs showing significant differences in abundance between healthy and diseased trees in root, needle and soil. These informations, however, are discordant with respect to those reported in Tables S2 and S3. For example, in roots Burkholderiales appeared to be significantly different in Table S3 but they are absent in Table 2. The same applies for Sphingomonadales and Sphingomonas, etc.
Response 7: The reviwer is right that Table 2 listed the OTUS showing differences in abundance between healthy and disease trees. However, Table S2 and S3 listed phyla, order and genus, which differed in abundance between healthy and disease trees. The value in Table S2 and Table S3 were abundance sum of individual OTUs belonging to the same taxonomy. In that sensen, Table 2 should differ from Table S2 and S3. For example, in Table 2, the individual Otu000485 and Otu000403 in the root were all classified as Burkholderia, which showed differences; In Table S3, the abundance of Burkholderia was the sum of all OTUs belonging to Burkholderia, in which some of the OTUs showed difference in abundance between healthy and diseased trees, and some did not.
Minor points:
Line 49: delete “Soil”
Response : Deleted L49
Line 63: change “is” to “was”
Response : Change L68
Line 301: delete “-stain”
Response : Deleted L305
Reviewer 2 Report
The authors of this manuscript have carried out a considerable amount of work and have generated interesting results on the effect of natural infections by the Pine Wilt Disease (PWD) and the nematode Bursaphelenchus xylophilus to the bacterial community structure of Pinus thunbergii. The methods used are properly executed and adequately described in the manuscript, whereas the results are analyzed in a proper way. Some specific comments (e.g. editorial comments etc.) on the manuscript are provided on the attached annotated pdf file. In conclusion, according to my opinion, the present manuscript can be accepted for publication in Microorganisms after a minor revision of the manuscript in which the authors will address the abovementioned comments.

Author Response
Response to Reviewer 2 Comments
The authors of this manuscript have carried out a considerable amount of work and have generated interesting results on the effect of natural infections by the Pine Wilt Disease (PWD) and the nematode Bursaphelenchus xylophilus to the bacterial community structure of Pinus thunbergii. The methods used are properly executed and adequately described in the manuscript, whereas the results are analyzed in a proper way. Some specific comments (e.g. editorial comments etc.) on the manuscript are provided on the attached annotated pdf file. In conclusion, according to my opinion, the present manuscript can be accepted for publication in Microorganisms after a minor revision of the manuscript in which the authors will address the abovementioned comments.
We appreciate the valuable comments to improve the manuscript and have revised the manuscript thoroughly according to the comments.
Point 1: Be more precise! What do you mean "newly killed trees"?
Response 1: The “newly killed trees’ mean that the trees were infected with B. xylophilus in March and killed in Septermber, which were separated from the trees killed in the previous year (L79-80). We have revised this in the text as: trees infected by B. xylophilus in March and killed in Septermber.
Point 2: Better round to one decimal place.
Response 2: We chose to keep two decimal places because keeping one would cause the standard deviation in some indices to fail to show the value of a decimal. Such as the standard deviation is 0.04 in the SWC index, the valid value cannot be presented after retaining one significant digit (L165-167). We thought it is better to keep two decimals.
Point 3: Instead of letters, better use asterisks to show statistically significant differences between two means.
Response 3: Thank you for the suggestions. We have changed the letters to asterisks to show the significant differences in the table (L165-167).
Point 4: Give some explanations, why you made this hypothesis (We expected that the pine wilt disease will have a profound impact on the host bacterial community structure.). Maybe some examples of other diseases that affected the host bacterial community structure.
Response 4: Thank you for the constructive comments. We have added explanations of the hypothesis in this revision (L256-258) as follow: 1) The decline in plant fitness (e.g. healthy status) or changes in growth conditions caused by the host pathogens can affect the microbial community in leaves and roots of the host; 2) Previous studies have shown that B. xylophilus infection can change the diversity and structure of endophytic wood-colonizing bacteria of P. pinaster trees.
Minor points:
L12 “forest” add “s”
Response: Added L12
L12 change “the study” to “information”
Response: Changed L12
L13 change “uncommon” to “limited”
Response: Changed L13
L24 change “had effect on” to “affected the”
Response: Changed L25
L56 “B.xylophilus” After point leave a space
Response: Left L56
L57 deleted “study of the”
Response: The original text expression has been modified. (L56-63)
L59 “relationship” add “s”
Response: Added L64
L66 change “study” to “Study”
Response: Changed L71
L67 “Nanjing” behind add “, China”
Response: Added L72-73
L69 “oC” behind add “,”
Response: Added L74
L71 change “at” to “around”
Response: Changed L76
L84 “soil” behind add “surface”
Response: Added L89
L160 change “soli” to “soil”
Response: Changed L165
L161 change “were” to “are”
Response: Changed L166
L161 change “mean” to “means”
Response: Changed L166
Round 2
Reviewer 1 Report
The Authors have satisfactorily addressed my questions and the quality of the manuscript has been improved.